# Predicting and Testing Bioavailability of Magnesium Supplements

**DOI:** 10.3390/nu11071663

**Published:** 2019-07-20

**Authors:** Laura Blancquaert, Chris Vervaet, Wim Derave

**Affiliations:** 1Department of Movement and Sports Sciences, Ghent University, 9000 Ghent, Belgium; 2Laboratory of Pharmaceutical Technology, Ghent University, 9000 Ghent, Belgium

**Keywords:** Magnesium, supplements, bioaccessibility, bioavailability

## Abstract

Despite the presumption of the beneficial effects of magnesium supplementation, little is known about the pharmacokinetics of different magnesium formulations. We aimed to investigate the value of two in vitro approaches to predict bioavailability of magnesium and to validate this in subsequent in vivo testing. In vitro assessment of 15 commercially available magnesium formulations was performed by means of a Simulator of the Human Intestinal Microbial Ecosystem (SHIME^®^) and by dissolution tests. Two magnesium formulations with contrasting bioavailability prediction from both in vitro tests (best vs. worst) were selected for in vivo testing in 30 subjects. In vivo bioavailability was compared following one acute ingestion by monitoring blood magnesium concentrations up to 6 h following intake. The in vitro tests showed a very wide variation in absorption and dissolution of the 15 magnesium products. In the in vivo testing, a significant different serum magnesium absorption profile was found up to 4 h following supplement ingestion for the two supplements with opposing in vitro test results. Moreover, maximal serum magnesium increase and total area under the curve were significantly different for both supplements (+6.2% vs. +4.6% and 6.87 vs. 0.31 mM.min, respectively). Collectively, poor bioaccessibility and bioavailability in the SHIME model clearly translated into poor dissolution and poor bioavailability in vivo. This provides a valid methodology for the prediction of in vivo bioavailability and effectiveness of micronutrients by specific in vitro approaches.

## 1. Introduction

Magnesium (Mg^2+^) is an essential mineral involved in numerous metabolic processes, thereby playing an important role in the physiological function of the brain, heart, and skeletal muscle [1,2]. Mg^2+^ is a cofactor in the activation of hundreds of enzymatic processes regulating diverse biochemical reactions, including energy metabolism, protein synthesis, muscle and nerve function, blood glucose, and blood pressure control [3]. Magnesium is primarily stored in bone, muscle, and soft tissue, and less than 1% is present in the extracellular fluid [4]. Roughly 30% of ingested magnesium through food or drinking water is absorbed by the intestine, although the extent of absorption depends on the body magnesium status (increased in case of Mg^2+^ deficiency). Magnesium homeostasis is further regulated through the secretion and reabsorption in the kidneys, where about 95% of the filtered magnesium is reabsorbed. Transfer of magnesium from serum to urine begins immediately when magnesium pools are saturated.

Several dietary surveys and epidemiologic studies performed in the USA and EU, revealed that on average people have an intake of dietary magnesium lower than the Recommended Daily Allowance (RDA) of 320 to 420 mg/day [5,6]. This is most likely a result of increased consumption of processed foods. Based on these findings, it is often suggested that over 50% of the normal population may have marginal magnesium deficiencies [7,8], although it should be noted that intake of dietary magnesium greatly differs between different countries and between different subpopulations (young vs. old, sedentary vs. athlete) [9,10]. Imbalance in magnesium levels in the body (hypomagnesemia) might result in unwanted neuromuscular, cardiac or nervous disorders leading to symptoms such as neuromuscular dysfunction, muscle weakness, and muscle cramping. The maintenance of adequate body magnesium levels is, therefore, very important for optimal physical performance and post-exercise recovery [1,11,12].

However, caution should be warranted when drawing conclusions concerning magnesium deficiency based on the RDA without the direct measurement of magnesium status. However, up until now, there are no readily available and easy methods to assess magnesium status. Serum magnesium analysis is still the most widely available method as it appears to be clinically the most practical, accessible, and expeditious method for identifying changes in magnesium homeostasis [13]. However, it is no ideal method since total serum magnesium concentration is not necessarily a good reflection of total body magnesium status [14], but it can be of value for determining acute changes in the intake or excretion of magnesium [13].

As magnesium plays an important role in physiological processes relevant to both health and sports performance, it is nowadays a popular nutritional supplement, with self-reported dietary magnesium use ranging around 7% in general population [15], and up to 25% in athletic populations [16]. Towards health, supplementation was already demonstrated to be beneficial in the treatment of, among others, preeclampsia, migraine, depression, coronary artery disease, and asthma [2]. However, research has been equivocal in regard to magnesium supplementation’s effect on exercise performance. Although some studies have been supportive, others found no benefit to exercise performance, as reviewed by Finstad et al. [1]. Moreover, in their meta-analysis, Garrison et al. [17] conclude that it is unlikely that magnesium supplementation provides a clinically meaningful treatment for skeletal muscle cramps. It is therefore not clear whether magnesium supplementation, beyond the maintenance of an adequate dietary intake of the mineral, is effective in enhancing performance and recovery from exercise.

It should be considered, however, that potential effects of magnesium supplementation may not yet be fully uncovered due to methodological problems. Suboptimal supplement effectiveness may be one such problem as this often differs between studies. Magnesium supplements are available in a variety of forms. Conventional magnesium supplements that can be found on the market nowadays contain one of the two distinct sources of elemental magnesium, inorganic, or organic salts of magnesium [18]. Inorganic salts (e.g., Mg oxide) provide a high loading of elemental magnesium but exhibit a very limited bioavailability as a result of their poor solubility. The organic sources of magnesium, on the other hand, offer high levels of solubility, but provide limited levels of elemental magnesium (e.g., Mg citrate). Studies on the bioavailability of different magnesium salts consistently demonstrate that organic salts of magnesium (e.g., Mg citrate) have a higher bioavailability than inorganic salts (e.g., Mg oxide) [19]. This finding was also confirmed by a recent study in which both urinary excretion and serum levels of magnesium were significantly higher after single-dose administration of these two supplements in a randomized cross-over study design [20].

Similar to magnesium, a lot of other micronutrients currently on the market in the form of supplements may have suboptimal efficacy due to poor solubility and/or bioavailability. It might, therefore, be useful to provide a valid and reliable in vitro approach to predict these parameters, in order to assure the efficacy of micronutrient supplements in vivo. This could help to restrain the use of non-effective supplements and enhance the quality and efficacy of supplements on the market. In the current paper, 15 different magnesium formulations were tested in vitro by both a Simulator of the Human Intestinal Microbial Ecosystem (SHIME^®^) and by dissolution tests, respectively. Based on these results, two magnesium formulations with opposing in vitro results were selected for further in vivo bioavailability testing following one acute ingestion. The current paper thus aimed to investigate the value of two in vitro approaches to predict bioavailability of micronutrients in general and magnesium in particular and to compare the in vivo bioavailability of two magnesium supplements with opposing in vitro results by measuring serum magnesium levels and urinary magnesium excretion.

## 2. Materials and Methods

### 2.1. In Vitro Testing

#### 2.1.1. In Vitro SHIME Simulation

In vitro assessment of bioaccessibility and bioavailability of 15 different commercially available formulations of magnesium (Table 1 and Appendix A) was performed by means of a Simulator of the Human Intestinal Microbial Ecosystem (SHIME^®^, ProDigest, Belgium). The SHIME consists of a succession of five reactors simulating the different parts of the human gastrointestinal tract [21]. For this specific study, a TWINSHIME setup was adapted to simulate multiple series of two reactors simulating first the stomach and then the small intestine. The physiological conditions of the test were adapted from the Infogest consensus digestion method [22]. Briefly, in the stomach, either a defined amount of SHIME nutritional medium (in g/L: Arabinogalactan (1.2), pectin (2), xylan (0.5), starch (4), glucose (0.4), yeast extract (3), peptone (1), mucin (3), l-cystein-HCl (0.5)), simulating food intake, or a fasted stomach solution is added, depending on whether the experiment is performed under fed or fasted conditions, respectively. Fed conditions were simulated under stirring conditions during 2 h at 37 °C, with a sigmoidal decrease of the pH profile from 4.7 to 2.0. Under fasted conditions, the pH was maintained stable at 2 for 45 min. After the stomach, a standardized pancreatic and bile liquid was added to the reactor for the simulation of small intestinal conditions [22,23]. In addition, simulation of the absorption processes was conducted by means of a static dialysis procedure with a cellulose membrane (cut-off = 14 kDa). By introducing the small intestinal suspension within a dialysis membrane, small molecules, such as digested amino acids, sugars, micronutrients, and other small molecules, are gradually removed from the upper gastro-intestinal matrices. Samples were collected from the fed stomach at 45 and 120 min, from the fasted stomach at 45 min. Sampling of both intestinal content (bioaccessible fraction) and dialysis solution (absorbed fraction) was done at 30, 60, 90, 120 and 180 min.

#### 2.1.2. In Vitro Dissolution Testing

The release rate of magnesium of the same 15 formulations (tablets or capsules) containing different magnesium salts was also determined using the USP paddle method (VK 7010, Vankel, Cary, NC, USA) in 900 mL dissolution medium (0.1N HCl and phosphate buffer pH 6.8). The temperature of the dissolution medium was maintained at 37 ± 0.5 °C, while the rotation speed was set at 100 rpm. Sinkers were used to prevent floating of the capsule formulations during dissolution testing. Samples of 2 mL were withdrawn after 10, 30, 60, 90, and 120 min. From the drug release profile, the time point of 80% drug release was determined.

#### 2.1.3. Determination of Magnesium Levels

For both in vitro tests, the concentration of magnesium was measured with a magnesium assay kit (Sigma-Aldrich, Belgium). The magnesium concentration is determined by a coupled enzyme assay that takes advantage of the specific requirement of glycerol kinase for Mg^2+^, resulting in a colorimetric (450 nm) product proportional to the Mg^2+^ present. The assay gives a linear range of 3–15 nmoles (0–7.3 ng/µL in 50 µL) of Mg^2+^ and exhibits no detectable interference with Fe^2+^, Cu^2+^, Ni^2+^, Zn^2+^, Co^2+^, Ca^2+^, and Mn^2+^. A positive control (5 ng Mg/µL) was included in each assay.

### 2.2. In Vivo Bioavailability

Aim: Two laboratory studies (phase A and phase B) have been performed to evaluate in vivo bioavailability. The aim of the phase A study was to define the best biological sampling strategy and analytical method to monitor magnesium absorption following an acute single ingestion through blood and urine analysis. For this phase, one supplement with presumably good bioavailability, as predicted from both in vitro tests was selected (Ultractive magnesium (A)) and compared to a placebo. This way, the in vivo monitoring could be tested on a magnesium formulation that is considered to be well absorbed. Supplement A was chosen because it showed the best overall results in the in vitro SHIME and dissolution testing. In a follow-up phase B study and based on the results obtained in phase A, the aim was to compare the bioavailability of two supplements with opposing bioavailability profile (best and worst), as predicted from both in vitro tests. Therefore, bioavailability following one acute ingestion of either Ultractive magnesium (A) or B-magnum (O) was tested in vivo.

Supplements: Ultractive magnesium contains both organic and inorganic Mg^2+^ salts. One tablet consists of 248.72 mg Mg oxide (containing 60% Mg, corresponding to 149 mg Mg) and 380.72 mg Mg glycerophosphate (containing 12.37% Mg, corresponding to 47 mg Mg). In total, one tablet thus contains 196 mg elementary Mg. B-magnum contains 750 mg Mg oxide (containing 60% Mg), corresponding to 450 mg Mg. The placebo supplement is composed of 400 mg glycine, 300 mg hydroxypropylmethylcellulose, 500 mg cellulose, 6 mg silicium dioxide, and 1 mg magnesium stearate.

Subjects: For both phases of the in vivo testing, different subjects were recruited. All subjects were in good health and non-specifically trained, but some of them took part in some form of recreational exercise one to three times per week. The inclusion criterion was age from 18–50 years. Exclusion criteria were smoking and using dietary supplements (including but not restricted to vitamins and minerals) in the last three months preceding and during the study.

Study Protocol: Subjects were asked (1) to refrain from heavy exercise on the evening before a test day, as well as during the test day, (2) to restrain from taking nourishments with a high magnesium content on the evening before each test day (e.g., nuts, wholegrain products, beans, green vegetables, fish, banana, chocolate), (3) to eat the same meal before each test day and to consume no alcohol the day before and during the test days. On the test days, subjects arrived in a fasted condition and received standardized meals and only drank magnesium-free water. Different test days were separated by at least one week. All subjects gave their informed consent for the described experiments and the studies were approved by the Local Ethics Committee (Ghent University Hospital).

#### 2.2.1. Phase A

Subjects: Ten healthy men volunteered to participate in this study. The subjects’ age, weight, height, and body mass index (BMI) were 24.8 ± 8.0 years, 75.7 ± 9.7 kg and 179.1 ± 5.2 cm and 23.5 ± 2.1 kg/m^2^, respectively.

Study protocol: A double-blind, placebo-controlled, randomized crossover design was used in this phase. The study consisted of two test days, during which subjects ingested either two tablets of Ultractive (A) magnesium (corresponding with 392 mg elementary magnesium) or two placebo tablets. The choice for two tablets was made to ensure that a possible increase in magnesium levels could be detected. At arrival, a catheter was inserted in an antecubital vein, and a first fasted venous blood sample was collected (serum gel tube). Hereafter, a standardized breakfast low in magnesium content (120 g of white bread and 56 g of strawberry jam) was consumed. One hour later, a second venous blood sample was collected before the subjects ingested the supplement (two tablets of Ultractive (A) magnesium or two placebo tablets). Consecutively, eight blood samples were withdrawn from the antecubital vein at 30, 60, 90, 120, 150, 180, 240, and 360 min following ingestion of the supplements. Serum tubes were left at room temperature for 30 min to obtain complete coagulation, followed by 5 min of centrifugation at 16,000 g. Serum was then frozen at −20 °C until subsequent analysis. Urine was collected in a container during the first 6 h following supplement intake, and in the subsequent 18 h in another container. Urine aliquots were collected from each container and frozen at −20 °C until subsequent analysis. Immediately after the blood sample at 180 min, subjects received a standardized lunch (white sandwich with cheese, tomato, cucumber, carrots, salad, and mayonnaise). During the test days, subjects were allowed to drink magnesium-free water ad libitum.

#### 2.2.2. Phase B

Subjects: Fifteen healthy male and 15 healthy female subjects volunteered to participate in this study. The subjects’ age, weight, height, and BMI were 24.7 ± 6.8 year, 75.5 ± 9.1 kg and 181.8 ± 6.2 cm and 22.8 ± 2.5 kg/m^2^ for men and 23.1 ± 2.8 year, 64.3 ± 6.0 kg and 170.6 ± 7.0 cm and 22.1 ± 1.9 kg/m^2^; for women, respectively.

Study protocol: A randomized crossover design was used in this phase. The study consisted of three test days, during which subjects ingested either one or two tablets of Ultractive (A) magnesium (corresponding to 196 mg and 392 mg elementary magnesium, respectively) or one tablet of B-magnum (O) (450 mg elementary magnesium). Because the manufacturer prescribed daily intake of one tablet of Ultractive (A), this condition was incorporated in the study. However, because the amount of elementary magnesium differs largely between one tablet of each supplement and because two tablets of Ultractive (A) were tested in the first phase, this condition was also included. The number of ingested tablets thus was not the same on the different test days, and the Ultractive (A) and B-magnum (O) tablets looked slightly different, making blinding of supplement intake impossible. The course of the test days was exactly the same as described for phase A, except for the supplements that were ingested.

Determination of serum and urinary magnesium levels: Determination of serum and urinary magnesium was measured with the Alinity C Magnesium Reagent Kit on an Abbott Alinity C device (Abbott, Wiesbaden, Germany), as prescribed by the manufacturer. In short, the method is based on the principle that magnesium, present in the samples, is a cofactor in an enzymatic reaction with isocitrate dehydrogenase. The rate of increase in absorbance at 340 nm, due to the formation of NADPH, is directly proportional to the magnesium concentration.

Statistics: Statistical analyses of the in vivo testing were performed using SPSS 25.0 software (SPSS, Chicago, IL, USA). Values are presented as mean ± SD, and statistical significance threshold was set at *p* ≤ 0.05. A repeated measures ANOVA with two within factors (time and supplement) was used to compare different measurements between the test days of both in vivo phase A and B. In case of a significant interaction effect, pairwise comparisons were made for each time point separately between two conditions and for each time point of each condition compared to pre-supplement intake (time point 0). For the serum magnesium levels, measured on 10 time points, Bonferroni correction was applied for the pairwise comparisons (level of significance is *p* ≤ 0.005 instead of *p* ≤ 0.05). For phase A, paired-samples T-tests were used to compare other parameters (maximal serum magnesium increase, incremental area under the curve) between the two testing days. For phase B, repeated-measures ANOVA with one within factor (condition) were used to compare other parameters (maximal serum magnesium increase, incremental area under the curve) between the three testing days.

## 3. Results

### 3.1. In Vitro Testing

#### 3.1.1. In Vitro SHIME Simulation

The SHIME^®^ technology was applied to simulate the behavior of 15 different Mg^2+^ formulations in the stomach and small intestine under fasted and fed conditions (Table 1). Total Mg^2+^ content (mg) was measured upon dosing the content of the stomach, and small intestinal compartment in either simulated fasted or fed conditions. By taking into account the total amount of Mg^2+^, present in each formulation, the proportional Mg^2+^ release was calculated. Since Mg has its highest solubility in an acidic environment such as the stomach, this is described as bioaccessible Mg. However, Mg is mainly absorbed in the distal part of the small intestine, and this is referred to as the bioavailable Mg.

The Mg^2+^ release from the different formulations in the stomach (Figure 1), reveals three main response scenarios: (1) Complete or very high release of Mg^2+^ under both fasted and fed conditions: Ultractive Magnesium (A), Magné Vie B6 (B), Polase (H) and High Absorption Magnesium (J), (2) complete release of Mg^2+^ under fed conditions with a lower release under fasted conditions: Mag 2 (D), (3) no or limited release of Mg^2+^: Magnerot (F), Magnesium Citrate 200 mg (M), MagOx 400 (K), Biolectra (I), Magnesium 500 mg (L), Mag 2 24 (G), Promagnor (C), B-Magnum (O), Slow-Mag (N), Magnesium Verla (E) (in order from intermediate to no release).

Moreover, the absorption efficiencies in the small intestine (% Mg^2+^ absorbed versus initial dose) was distinctly different for the 15 formulations (Figure 2), revealing that: (1) Magnerot (F), Polase (H), Ultractive Magnesium (A), Magné Vie B6 (B) and High Absorption Magnesium (J) resulted in an efficient absorption of Mg^2+^ under fasted/fed conditions, (2) Magnesium Verla (E) resulted in a moderate absorption under fed conditions while absorption efficiency under fasted conditions was similar to the best-performing products. (3) Slow-Mag (N) had a moderate absorption under both fed and fasted conditions, (4) Mag 2 (D) and Magnesium Citrate 200 mg (M) also had a moderate absorption efficiency, which was slightly higher under fed conditions (~higher solubility in stomach under fed conditions). (5) MagOx 400 (K), Biolectra (I), Magnesium 500 mg (L), Promagnor (C), Mag 2 24 h (G) and B-Magnum (O) resulted in the worst Mg^2+^ absorption efficiency (in order of decreasing Mg^2+^ release).

#### 3.1.2. In-Vitro Dissolution Testing

The release rate of magnesium from the dosage forms tested in this study ranged from immediate-release (with >80% of the magnesium content released within 10 min) to slow-release (>120 min to release 80% of the magnesium content) (Table 2). These differences in release rate are linked to the magnesium salts incorporated in the formulations (e.g., poorly soluble magnesium oxide vs. very soluble magnesium chloride) and/or to the excipients included in the dosage forms (e.g., Slow-Mag (N) provides sustained release of the very soluble magnesium chloride, while release of this salt from Polase (H) is fast). While the trade name of some of the formulations clearly indicates that a slow magnesium release is targeted (Mag 2 24 h (G), Slow-Mag (N)), the rationale for the slower release from other dosage forms is less evident. The impact of formulation composition on release is also evident from the magnesium oxide-containing formulations. Ultractive Magnesium (A) and MagOx 400 (K) have an immediate release in acid medium (*t*_80%_ < 10 min), while B-Magnum (O) has intermediate release rate (*t*_80%_ of 60 min) and Promagnor (C), Biolectra (I) and Magnesium 500 mg (L) a slow release rate, respectively.

When linking the results of the two in vitro approaches (SHIME and in vitro dissolution testing), there is a strong correlation between both the Mg bioaccessible (end stomach) in fed conditions and the percentage Mg released in pH 6.8 (*r* = 0.966, *p* < 0.001, Figure 3A) on the one hand, and between the Mg bioavailable (end small intestine) in fed conditions and the percentage Mg released in pH 6.8 (*r* = 0.867, *p* < 0.001, Figure 3B). Similar strong correlations were found for the bioaccessible or bioavailable Mg in fasted conditions and the percentage Mg released in 0.1N HCl. Taken together the results on the in vitro experiments, magnesium supplements partially or completely consisting of organic magnesium formulation such as Supplement A, B, F, H, and J show the highest rate of both bioaccessibility, bioavailability and dissolution, although this was not found for the organic magnesium Supplements E and M, indicating that the magnesium formulation is not the only determining factor.

### 3.2. In Vivo Bioavailability

#### 3.2.1. Phase A

In the first phase of in vivo testing, one supplement with presumably good bioavailability, as predicted from both in vitro tests (Supplement A), was monitored through repetitive venous blood sampling following an acute single ingestion and compared to placebo on two separate test days. There was good agreement between baseline serum magnesium levels on the two test days (*r* = 0.716, *p* = 0.02). No significant interaction effect was present (*p* = 0.129) for the serum magnesium levels following two tablets of Supplement A (392 mg elementary magnesium) or placebo ingestion (Figure 4). When comparing the maximal serum magnesium increase (difference between maximal value and pre-supplement value), a significant difference was found between the two supplement conditions (Figure 5A). The maximal increase was 0.041 mmol/L following placebo ingestion (corresponds with 4.8%) and 0.071 mmol/L following two tablets of Supplement A (corresponds with 8.4%) (*p* = 0.05). Based on the profile of serum magnesium, total incremental area under the curve (iAUC) was calculated for the timeframe during which serum was collected, demonstrating a significant difference in iAUC between the placebo and Supplement A condition (placebo 4.25 mM.min vs. Supplement A 11.89 mM.min, *p* = 0.02) (Figure 5B).

No difference was found in urinary magnesium excretion in the first six hours following supplement ingestion (placebo: 0.23 mmol/h, two tablets of Supplement A: 0.22 mmol/h), neither in the subsequent 18 h (placebo: 0.17 mmol/h, two tablets of Supplement A: 0.20 mmol/h).

#### 3.2.2. Phase B

In the second phase of in vivo testing, the bioavailability following one acute ingestion of two supplements with opposing bioavailability profile (best and worst), as predicted from both in vitro tests, was compared by monitoring blood analysis, as the phase A demonstrated this to give a valuable representation of bioavailability. Therefore, bioavailability following one acute ingestion of either one or two tablets of Supplement A (corresponding to 196 mg and 392 mg elementary magnesium, respectively) or one tablet of Supplement O (450 mg elementary magnesium) was monitored. Figure 6A represents the absorption profile of one tablet of Supplement A vs. one tablet of Supplement O. A significant interaction effect was present (*p* = 0.013), and significant different serum magnesium values between both conditions were found at 30, 90, 150, 180 min and four hours following supplement ingestion (*p* < 0.005, Bonferroni correction for pairwise comparison of 10 time points). Furthermore, significant higher serum magnesium levels compared to pre-supplement levels (time point 0) were found following one tablet of supplement A at 120, 150, 180 min and four hours following ingestion, but not at any time point following one tablet of Supplement O (*p* < 0.005, significant differences vs. time point 0 are not depicted in Figure 6A). Figure 6B represents the absorption profile of two tablets of Supplement A vs. one tablet of Supplement O, for which the interaction effect was also significant (*p* = 0.032). Significantly different serum magnesium values between both conditions were found at 120, 150, 180 min and four and six hours following supplement ingestion (*p* < 0.005, Bonferroni correction for pairwise comparison of 10 time points). In accordance, significantly higher serum magnesium levels compared to pre-supplement levels (time point 0) were found following two tablets of Supplement A at 120, 150, 180 min and four and six hours following ingestion (*p* < 0.005, significant differences vs. time point 0 are not depicted in Figure 6B). No significant interaction effect was present between the absorption profile of one and two tablets of Supplement A (*p* = 0.179).

When comparing the maximal serum Mg increase (difference between maximal value and pre-supplement value), a significant main effect of condition was found (*p* = 0.005). Pairwise comparisons revealed significant differences between one tablet of Supplement A and one tablet of Supplement O (*p* = 0.048) and between two tablets of Supplement A and one tablet of Supplement O (*p* = 0.001) (Figure 7A). The maximal increase was 0.050 mmol/L (+6.2%) and 0.062 mmol/L (+8.0%) following ingestion of one or two tablets of Supplement A, respectively, and 0.034 mmol/L (+4.6%) following ingestion of one tablet of Supplement O. When considering the timeframe from supplement intake till 360 min following ingestion, a significant main effect of condition was found for the total area under the curve (*p* = 0.021). As depicted in Figure 7B, a significant difference is found in iAUC between one tablet of Supplement A and one tablet of Supplement O (6.87 mM.min vs. 0.31 mM.min, respectively, *p* = 0.009) and between two tablets of Supplement A and one tablet of Supplement O (6.80 mM.min vs. 0.31 mM.min, respectively, *p* = 0.011).

No difference was found in urinary magnesium excretion in the first six hours following supplement ingestion (one tablet of Supplement A: 0.17 mmol/h, two tablets of Supplement A: 0.16 mmol/h, one tablet of Supplement O: 0.17 mmol/h), neither in the subsequent 18 h (one tablet of Supplement A: 0.18 mmol/h, two tablets of Supplement A: 0.19 mmol/h, one tablet of Supplement O: 0.18 mmol/h).

## 4. Discussion

The aim of the current study was to investigate the value of two in vitro approaches to predict the bioavailability of magnesium supplement ingestion and validate their predictive power by actual in vivo testing. Based on the in vitro approaches, two supplements with the most divergent results concerning bioaccessibility and bioavailability were selected for in vivo testing. Importantly, poor performance in the in vitro tests was shown to translate into poor bioavailability in vivo.

Two different in vitro approaches were used to compare the commercial products. The SHIME model is designed to well resemble the human gastro-intestinal tract [21], whereas the dissolution tests assessed relatively simple solubility characteristics of the products. The in vitro gastro-intestinal simulator (SHIME) experiments and in vitro dissolution tests showed a very wide variation in dissolution and absorption of the 15 tested commercially available magnesium products. Moreover, poor bioaccessibility and bioavailability in the SHIME model clearly translate into poor dissolution, as indicated by the strong correlation between the results of both approaches. It was anticipated that supplements that perform poorly on the two in vitro tests would also perform poorly on the in vivo bio-availability testing.

In order to allow comparison between the in vitro testing and the in vivo bioavailability, a good approach for the latter first needed to be established. Good validity of the in vivo approach of using the measurement of serum magnesium levels 6 h following supplement ingestion was shown in phase A. Baseline serum magnesium concentrations were within the range of 0.7 and 1.1 mM, which is described in the literature as a normal range in healthy people [2,24]. Moreover, reproducible baseline serum magnesium levels were found between different test days. The serum levels of magnesium express a circadian variation of around 6% [19,25]. This is confirmed in phase A of the current in vivo trial after placebo supplement ingestion, for which an average maximal increase of 4.8% (maximal value compared to the baseline value) was found throughout the timeframe of serum collection. However, following the ingestion of the supplement with the best-predicted bioavailability based on in vitro test, the average maximal increase was 8.4% in phase A of the in vivo trial, thus overruling this circadian variation. This is in accordance with different recent studies that directly compare the bioavailability of organic and inorganic magnesium salts and demonstrate significant differences in serum magnesium levels upon a single oral dose [20,26,27]. Based on this, it was concluded that 6 h serum magnesium measurement following an acute single ingestion was a valid approach for comparing the in vivo bioavailability of two supplements in phase B.

For the magnesium supplement with the worst bioavailability profile, based on both in vitro approaches (Supplement O), the non-significant increase in serum magnesium levels in the in vivo phase B trial were in the same range as those found following placebo ingestion (average maximal increase of 4.6% compared to baseline), and are thus smaller than the circadian variation reported in literature. However, in accordance with the results of phase A, the supplement with the best bioavailability profile based on the in vitro results (supplement A) demonstrates higher bioavailability, overruling this circadian variation and leading to actual higher serum magnesium levels. Accordingly, in phase B of the current study, a maximal percentage increase in serum magnesium levels were 6.2% and 8.0% following ingestion of either one or two tablets of the supplement with the best bioavailability profile (Supplement A) based on the in vitro results, respectively.

It is known that serum magnesium values reflect only 1% of the body magnesium content [14] since the absorbed magnesium disappears quickly from the circulation and is distributed into body stores (bone, muscle, and soft tissues) and excreted renally. It is known that serum magnesium concentrations are therefore not necessarily a good reflection of total body magnesium status. The magnesium tolerance test, in which the percentage of magnesium retained after parenteral administration is determined, is often offered as an alternative [4]. Interestingly, many studies in the literature use 24 h urinary magnesium excretion as an estimate of bioavailability following supplement ingestion. This parameter was also tested in the current study, but no significant differences were found in urinary magnesium excretion in both phase one and two. Accordingly, the study of Walker et al. [26] found no statistically significant difference in bioavailability between both organic (Mg citrate) and inorganic (Mg oxide) magnesium preparations when comparing renally excreted magnesium after single-dose administration. However, the study of Kappeler et al. [20] and Lindberg et al. [28], in which the same supplements were used, did report a significant difference for this parameter. It should be noted that Kappeler et al. [20] included a five-day phase of magnesium supplementation for saturation of the study subjects’ Mg-pools before single-dose magnesium administration, which was not the case in the current study. Following saturation of the body stores as performed in Kappeler et al. [20], the absorbed magnesium is not stored but is excreted renally.

The present results further support the general pattern that inorganic magnesium compounds are not as easily absorbable as organic magnesium compounds by both in vitro and in vivo testing. As mentioned, for the in vivo tests, the current study tested two supplements with opposing bioavailability profile as predicted from both in vitro tests. The selected supplement which had one of the highest predicted bioavailability differs from other formulations as it contains both inorganic (Mg oxide) and organic (Mg glycerophosphate) magnesium salts, while the supplement with the worst predicted bioavailability consists of only inorganic magnesium (Mg Oxide). The disadvantage of organic sources of magnesium holds that, although they have a higher solubility, they provide only limited levels of elementary magnesium in contrast to inorganic salt which offers a high loading of elementary magnesium. Yet, in phase B of the current study, no additive effect of ingestion of two tablets of Supplement A (396 mg elementary magnesium) compared to one tablet (196 mg elementary magnesium) on serum magnesium levels was found. Moreover, ingestion of one tablet of Supplement A led to higher serum magnesium levels compared to one tablet of Supplement O, although the latter contains more than twice as much elementary magnesium (196 vs. 450 mg elementary magnesium, respectively). These findings suggest that the solubility of a magnesium supplement is of greater relevance for in vivo bioavailability than the loading (amount elemental magnesium). One important notice is that the tolerable upper intake levels (UL) for supplemental magnesium are set at 350 mg per day in the US and 250 mg per day in Europe. It should be noted, however, that some commercially available magnesium formulations tested in the current study contain higher amounts of elemental magnesium in one tablet (see Table 1). Thus, they theoretically exceed the UL—if absorption efficiency would be 100%—and do not comply with safety and health recommendations. As in vivo bioavailability of one tablet of Supplement A (196 mg magnesium) was not higher compared to ingestion of two tablets (392 mg magnesium), and the ingestion of two tablets exceeds the UL, ingestion of only one tablet should be recommended. For Supplement O, containing 450 mg magnesium per tablet, the UL is thus exceeded, but it is highly unlikely that this effectively led to an overdose, as no bioavailable magnesium could be measured in serum following one acute ingestion.

Based on the in vitro tests, some supplements containing organic magnesium formulations (such as Supplement B, F, H) show similar bioavailability and dissolution results compared to Supplement A, which contains both organic and inorganic magnesium. It should be noted that Supplements B, F and H contain lower magnesium content and that similar percentages of bioaccessible or bioavailable Mg, as reported in Figure 1 and Figure 2, thus do not necessarily correspond to similar bioaccessible or bioavailable amounts of magnesium, respectively. Furthermore, in the current study, no organic magnesium supplement was tested in vivo. It thus remains to be established whether in vivo bioavailability of organic magnesium formulations differs from supplements containing both organic and inorganic magnesium.

One limit of the current study is that serum magnesium levels were collected up to 6 h following supplement ingestion. Magnesium is primarily absorbed in the distal parts of the small intestine and in the colon. Although magnesium absorption is known to decrease from 4–5 h post-ingestion, it cannot fully be excluded that a minor part of the ingested magnesium might not yet be absorbed within the 6 h timeframe during which serum was collected.

The present study was conducted in both males and females, making the results transferable to the total population. To date, no evidence indicates that intestinal magnesium absorption is different between sexes, but estrogen is known to influence magnesium distribution and excretion [29]. Since the concentration of this hormone is higher in females, most studies on magnesium bioavailability are only conducted in men. However, in this study, no effect of sex was found on serum magnesium concentration following a single acute ingestion, which served as an indirect measure of magnesium absorption and bioavailability.

## 5. Conclusions

In conclusion, this paper documents markedly divergent performance of commercial magnesium supplements. The in vivo Mg bioavailability was not related to the Mg content of the supplements, but rather to the in vitro solubility and bioaccessibility. Our study provides a valid methodology to predict in vivo outcomes and effectiveness by specific and simple in vitro approaches. Moreover, monitoring of serum magnesium levels was found to be a valuable measure of in vivo bioavailability. The in vitro methodology is useful for the prediction of bioavailability of micronutrients in general and can be a helpful approach to assure proper supplement use in future intervention studies and to avoid non-efficient micronutrient supplements on the market.

## Figures and Tables

**Figure 1 nutrients-11-01663-f001:**
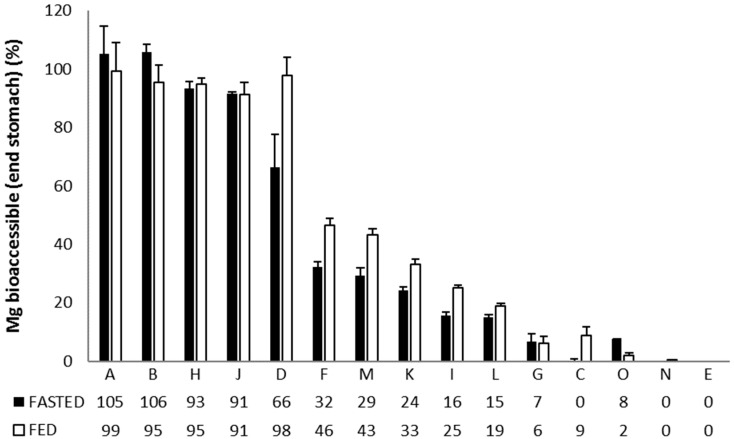
Relative (%) Mg^2+^ release during the stomach incubation upon simulated ingestion of 15 Mg-containing formulations under fasted and fed conditions. Data are shown as average (*n* = 3) ± standard deviation (STDEV).

**Figure 2 nutrients-11-01663-f002:**
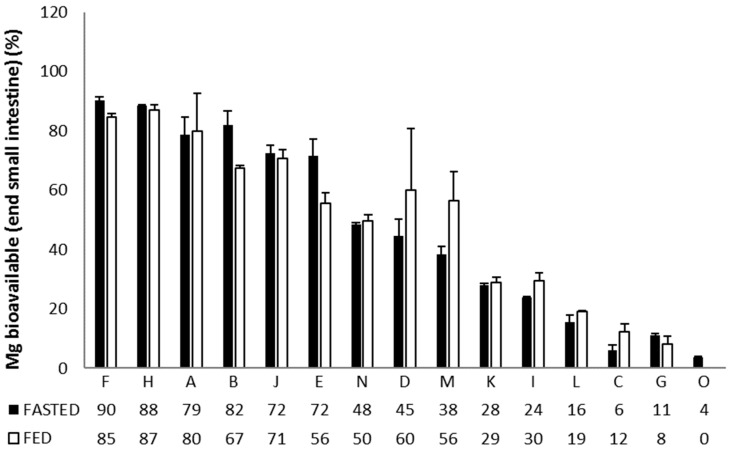
Relative (%) Mg^2+^ absorption during the small intestinal incubation upon simulated ingestion of 15 Mg-containing formulations under fasted and fed conditions. Data are shown as average (*n* = 3) ± STDEV.

**Figure 3 nutrients-11-01663-f003:**
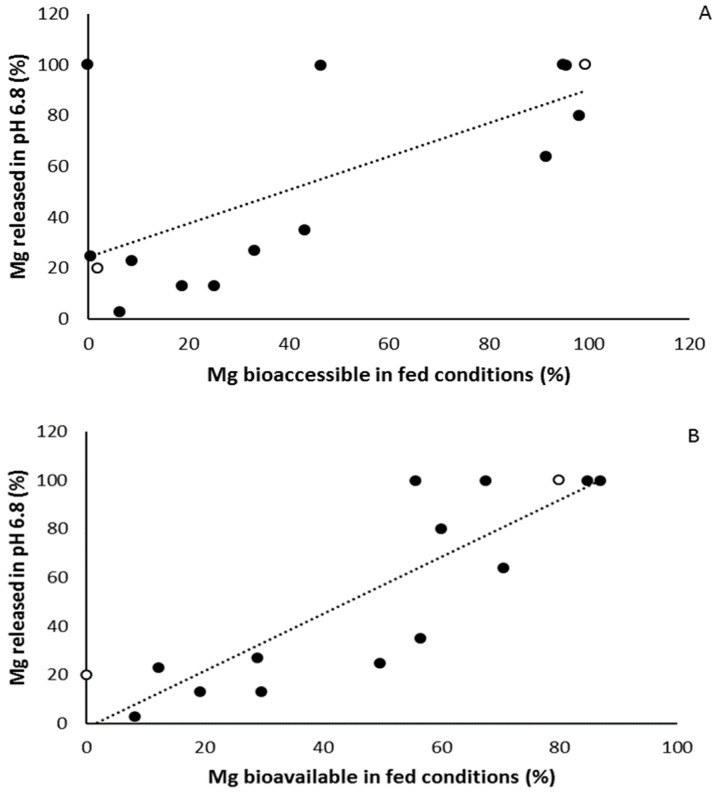
(**A**) Significant correlation between percentage Mg released in pH 6.8 (dissolution testing) and Mg bioaccessible in fed conditions (Simulator of the Human Intestinal Microbial Ecosystem (SHIME) testing). (**B**) Significant correlation between percentage Mg released in pH 6.8 (dissolution testing) and Mg bioavailable in fed conditions (SHIME testing). White dots represent the magnesium formulations that were further used for in vivo testing (Ultractive (A) and B-Magnum (O)), black dots are all other tested magnesium formulations.

**Figure 4 nutrients-11-01663-f004:**
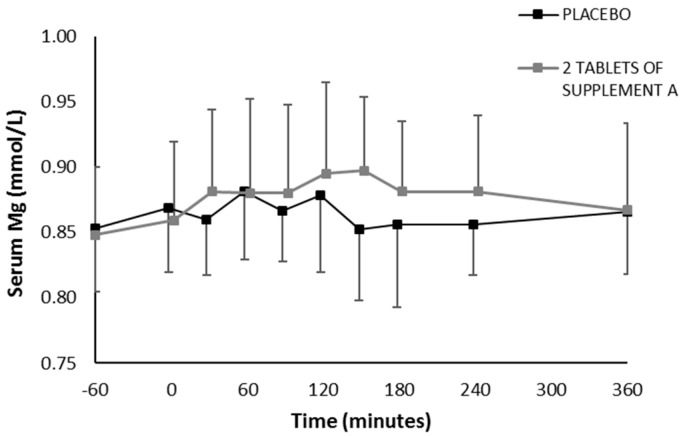
Serum magnesium levels following ingestion of two tablets of Supplement A or placebo. Concentrations were measured 60’ before and immediately before (0’) ingesting the supplements and 30’, 60’, 90’, 120’, 150’, 180’, 4 and 6 h after ingestion.

**Figure 5 nutrients-11-01663-f005:**
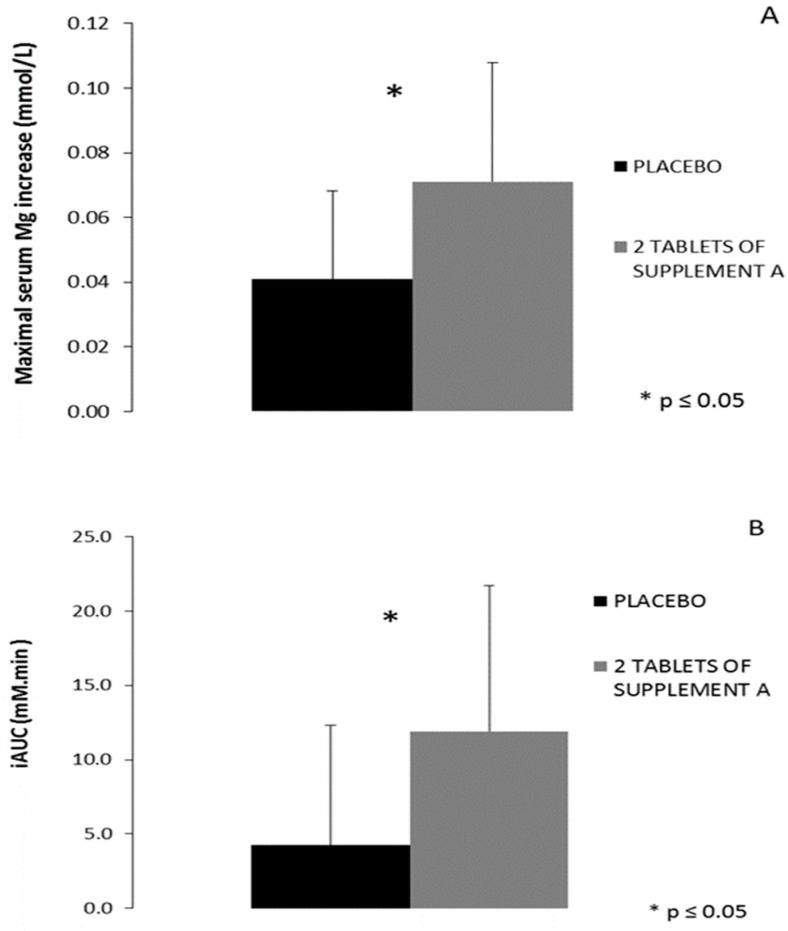
(**A**) The difference in serum magnesium concentration between the peak value and the concentration before ingestion of two tablets of either the placebo or Supplement A (* *p* = 0.05). (**B**) The incremental area under the curve (iAUC) of the serum magnesium concentration starting from 60’ before ingestion of the supplement or placebo (fasted sample) up to six hours after ingestion (* *p* = 0.02).

**Figure 6 nutrients-11-01663-f006:**
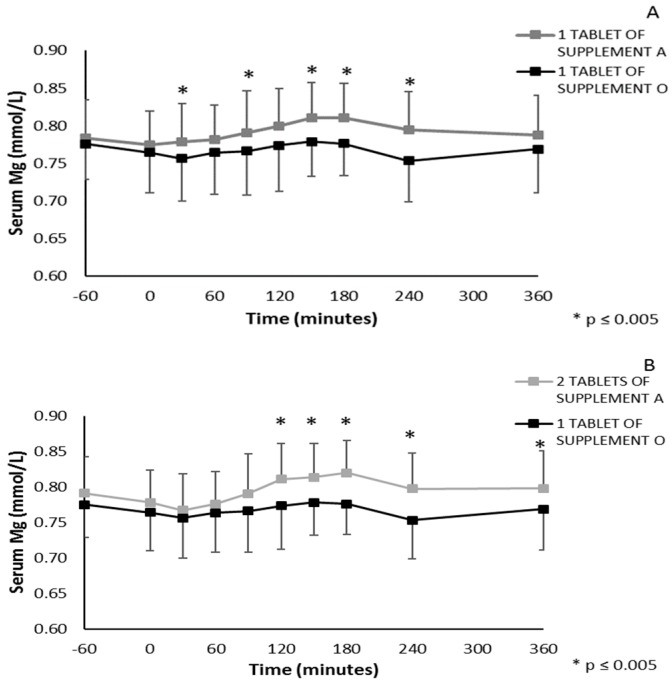
(**A)** Serum magnesium levels following ingestion of one tablet of Supplement A vs. one tablet of Supplement O, * *p* ≤ 0.005 (**B**) serum magnesium levels following ingestion of two tablets of Supplement A vs. one tablet of Supplement O. Concentrations were measured 60’ before and immediately before (0’) ingesting the supplement and 30’, 60’, 90’, 120’, 150’, 180’, 4 and 6 h after ingestion, * *p* ≤ 0.005.

**Figure 7 nutrients-11-01663-f007:**
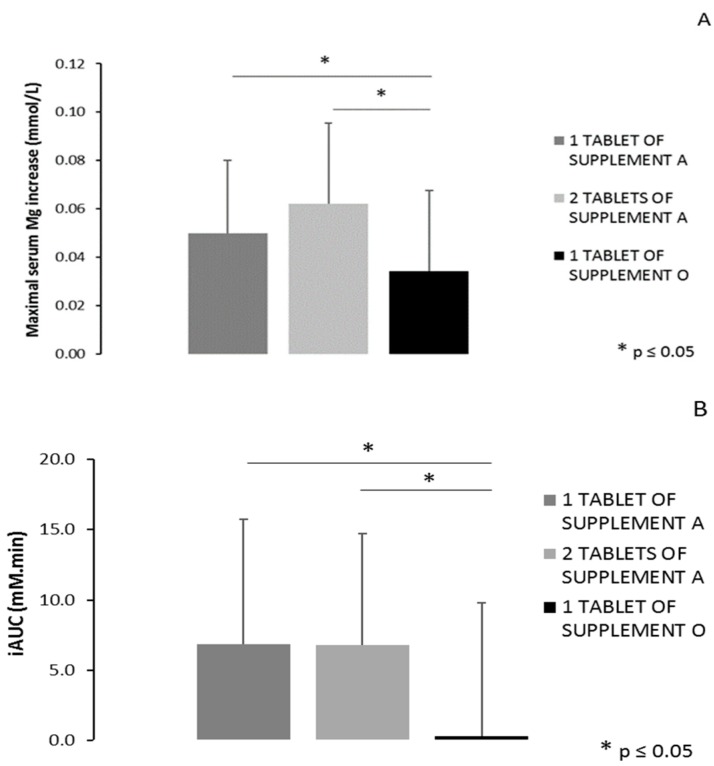
(**A**) The difference in serum magnesium concentration between the peak value and the concentration before ingestion of one or two tablets of Supplement A vs. one tablet of Supplement O (* *p* = 0.048 for one tablet of Supplement A vs. Supplement O and *p* = 0.001 for two tablets of Supplement A vs. Supplement O). (**B**) The incremental area under the curve (iAUC) of the serum magnesium concentration starting from ingestion of the supplement up to six hours after ingestion (* *p* = 0.009 for one tablet of Supplement A vs. Supplement O and *p* = 0.011 for two tablets of Supplement A vs. Supplement O). * *p* ≤ 0.05.

**Table 1 nutrients-11-01663-t001:** Overview of the 15 magnesium formulations used for the in vitro Simulator of the Human Intestinal Microbial Ecosystem (SHIME) and dissolution testing.

Code	Product	Manufacturer	Mg Content (mg)	Mg Salt	Pharmaceutical Form
A	Ultractive Magnesium	Oystershell	196	Mg oxide + Mg glycerophosphate	Tablet
B	MagnéVieB6	Sanofi-Aventis France	100	Mg citrate	Tablet
C	Promagnor	Merck	450	Mg oxide	Capsule
D	Mag 2	Cooper	100	Mg carbonate	Tablet
E	Magnesium Verla N tablets	Verla	40	Mg citrate + Mg bis (hydrogen-l-glutamate)	Tablet
F	Magnerot	Wörwag PHARMA	32.8	Mg orotate dihydrate	Tablet
G	Mag 2 24 h	Cooper	300	Mg oxide	Capsule
H	Polase	Pfizer	19	Mg citrate	Tablet
I	Biolectra	Hermes Arzneimittel	400	Mg oxide	Capsule
J	High Absorption Magnesium	Doctor’s Best	100	Mg glycinate lysinate chelate	Tablet
K	MagOx 400	Akorn Consumer Health	241.3	Mg oxide	Tablet
L	Magnesium 500 mg	Nature’s Bounty	500	Mg oxide	Tablet
M	Magnesium Citrate 200 mg	Now Foods	200	Mg citrate	Tablet
N	Slow-Mag	Purdue Products	71.5	Mg chloride	Tablet
O	B-Magnum	Merck	450	Mg oxide	Tablet

**Table 2 nutrients-11-01663-t002:** Dissolution rates expressed as time to release 80% of Mg content and % release of Mg after 120 min for the 15 different Mg formulations.

Code	Product	Time to Release 80% of Mg Content (min)		Mg Released (%) after 120 min	
		in 0.1N HCl	in pH 6.8	in 0.1N HCl	in pH 6.8
A	Ultractive Magnesium	<10	<10	95	100
B	MagnéVie B6	22	22	96	100
C	Promagnor	>120	>120	28	23
D	Mag 2	<10	120	98	80
E	Magnesium Verla N tablets	>120	43	4	100
F	Magnerot	10	<10	96	100
G	Mag 2 24 h	>120	>120	27	3
H	Polase	24	28	100	100
I	Biolectra	>120	>120	68	13
J	High Absorption Magnesium	37	>120	100	64
K	MagOx 400	<10	>120	100	27
L	Magnesium 500 mg	>120	>120	39	13
M	Magnesium Citrate 200 mg	>120	>120	64	35
N	Slow-Mag	>120	>120	0	25
O	B-Magnum	60	>120	95	20

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
