# Peer review of "Predicting and Testing Bioavailability of Magnesium Supplements"

_nutrients, 2019, doi:10.3390/nu11071663_

Reviewer 1 Report

“Predicting and testing bioavailability of magnesium 2 supplements” is an interesting study that examines the role of bioavailability of magnesium using in vitro and in vivo experiments. The authors show that the bioaccessibility and bioavailability of magnesium supplements is generally poor. The study addresses a major problem that has currently a lot of attention given the hype of magnesium supplementation. However, there are serious issues with the experimental design of the in vivo studies. Please find below my suggestions for further improvement of the manuscript.

1)     A major issue of the in vivo studies is the relatively short time following the ingestion of the supplements. Magnesium is mainly reabsorbed in the distal parts of the small intestine and in the colon. Given that gastric emptying may take up to 5 hrs, only a minimal amount of the supplements will have reached the intestinal segments where Mg is reabsorbed after 360 minutes. The authors therefore may therefore simply too early to measure the effects.

2)     A second issue is that participants were given a meal at 180 minutes during the testing period. This food intake will stimulate the release of insulin, which has a well known effect on cellular Mg uptake and renal Mg wasting. Indeed, in the uptake-curves (Figure 4 and 6), the magnesium levels are decreasing after 180 minutes, which is in line with this insulin effect. How can the authors exclude effects of the meal during their tests?

3)     The figures 4 and 6 give the average serum Mg concentration of all participants, which may mask potential small defects. Please provide individual data points of the participants.

4)     Why did the authors select compound A for in vivo testing, whereas compounds F and H have better bioavailability?

5)     The aim is unclear. The authors state that ‘the paper aimed to investigate the value of two in vitro approaches to predict bioavailability of micronutrients in general and magnesium in particular, and to validate their predictive power by actual in vivo testing.’. However, bioavailability is not measured in vivo, only serum Mg levels are measured, which is a very poor way to validate the in vitro bioavailability test. One would expect that the aim of this study was to compare which Mg supplement has the highest bioavailability, rather than validating the SHIME test. The in vivo experiments that would be required to validate the SHIME test (e.g. analysis of intestinal fluids after stomach/small intestine) are not part of this manuscript.

6)     The main conclusions are not supported by the data. The authors state that ‘Our study provides a valid methodology to predict in vivo outcomes and effectiveness by specific and simple in vitro approaches.’ It is completely unclear on which basis the authors draw this conclusion. Their in vitro studies showed >90% availability, whereas their in vivo studies showed no significant increase in serum Mg levels using mixed models (repeated ANOVA) analysis and only a minor increase using AUC. In my view, this shows that their in vitro tests poorly predicts in vivo outcomes.

7)     The discussion of the manuscript should be more extended. What do the authors propose? Which supplement should be used? Why? It is difficult to understand why a mix of organic/inorganic Mg would have a better bioavailability than organic mg salts.

Minor points

8)     The figure quality is rather poor. Better use professional software instead of excel for figure formatting.

9)     Please provide quantity units (+ dimension) in all axis labels e.g. Mg concentration (mmol/L)

10)  Harmonize fond size, axis labels etc. in figures (remove titles above the figure)

Author Response

We would like to thank the reviewer for his helpful and valuable suggestions and comments, which we feel have positively contributed to the revised version of the manuscript.

 “Predicting and testing bioavailability of magnesium 2 supplements” is an interesting study that examines the role of bioavailability of magnesium using in vitro and in vivo experiments. The authors show that the bioaccessibility and bioavailability of magnesium supplements is generally poor. The study addresses a major problem that has currently a lot of attention given the hype of magnesium supplementation. However, there are serious issues with the experimental design of the in vivo studies. Please find below my suggestions for further improvement of the manuscript.

 It is known that magnesium absorption starts around 1h following magnesium intake, and decreases at 4-5h following intake (Hardwick et al., 1990). Based on the available literature in which serum magnesium levels are used to measure magnesium bioavailability upon supplement ingestion, we decided to collect serum up to 6h after ingestion. Both in the studies of Dolberg et al (2017) and Kappeler et al (2017), maximal serum magnesium concentrations were reached at 2-4 hours following supplement intake. In the current article, we find the same range for time to maximal magnesium levels (3h following Supplement A), and at 6h post ingestion, serum magnesium levels are back at baseline, suggesting that the vast majority of magnesium is absorbed at this point. Together, these findings make us confident of the fact that most ingested magnesium was absorbed within 6h and we are thus not missing the effects. However, we agree with the reviewer that it might be possible that a small fraction of magnesium is not yet absorbed, and therefore addressed this in the discussion of the paper. Following sentence was added to the discussion (line 477-481):

‘One limit of the current study is that serum magnesium levels were collected up to 6h following supplement ingestion. Magnesium is primarily absorbed in the distal parts of the small intestine and in the colon. Although magnesium absorption is known to decrease from 4-5h post ingestion, it can not fully be excluded that a minor part of the ingested magnesium might not yet be absorbed within the 6h timeframe during which serum was collected.’

 Point 2: A second issue is that participants were given a meal at 180 minutes during the testing period. This food intake will stimulate the release of insulin, which has a well-known effect on cellular Mg uptake and renal Mg wasting. Indeed, in the uptake-curves (Figure 4 and 6), the magnesium levels are decreasing after 180 minutes, which is in line with this insulin effect. How can the authors exclude effects of the meal during their tests?

 We agree that the meal intake affects magnesium homeostasis. However, as we used a crossover design (every subject underwent every condition of supplement ingestion), the effect of meal intake is partly ‘ruled out’ since the exact same meal was given to every subject on every test day, and the insulin response is thus also expected to be in the same range within test days. Therefore, no different effect of meal intake is expected on the absorption of the different magnesium supplements. Furthermore, from a practical point of view, it was not evident to restrain subjects from meals for 6h. Moreover, when magnesium supplements are used in daily real-life setting, people also eat meals in the hours following supplement intake and the insulin response will thus also be present.

 Point 3: The figures 4 and 6 give the average serum Mg concentration of all participants, which may mask potential small defects. Please provide individual data points of the participants.

 We indeed checked for individual data points and no defects were present in these data, as can be seen on the figures below. We therefore chose to only represent mean values in figures 4 and 6, in order to improve the clarity of these figures.

 Phase A:

Individual data following placebo ingestion:

Individual data following ingestion of 2 tablets of supplement A:

Phase B:

Individual data following ingestion of 1 tablet of supplement A:

Individual data following ingestion of 1 tablet of supplement O:

Individual data following ingestion of 2 tablets of supplement A:

Point 4: Why did the authors select compound A for in vivo testing, whereas compounds F and H have better bioavailability?

Supplement A was the supplement with best ‘overall’ in vitro results. Both for the bioaccessible and bioavailbale (SHIME) measures, as well as for the dissolution results, supplement A belong to the three best performing supplements (highest relative absorption/release rates compared to the other supplements). Although Supplement F & H demonstrated good results on bioavailability (SHIME) and release rate, the results on bioaccesibility of the SHIME test were somewhat lower for these supplements. As we based our decision on all in vitro tests, Supplement A was selected for further in vivo testing. Following sentence was added to the method section to justify this choice (line 145-146): ‘Supplement A was chosen because it showed the best overall results in the in vitro SHIME and dissolution testing.’

Point 5: The aim is unclear. The authors state that ‘the paper aimed to investigate the value of two in vitro approaches to predict bioavailability of micronutrients in general and magnesium in particular, and to validate their predictive power by actual in vivo testing.’. However, bioavailability is not measured in vivo, only serum Mg levels are measured, which is a very poor way to validate the in vitro bioavailability test. One would expect that the aim of this study was to compare which Mg supplement has the highest bioavailability, rather than validating the SHIME test. The in vivo experiments that would be required to validate the SHIME test (e.g. analysis of intestinal fluids after stomach/small intestine) are not part of this manuscript.

We agree that serum magnesium levels are perhaps not the best way to validate the in vitro bioavailability test, but it provides an in vivo proxy/estimation of bioavailability that can be used on an adequatly powered study group to compare with in vitro tests. That said, we agree that the formulation of the aim of the study can be somewhat confusing. The aim was therefore reformulated to (line 94-98):

‘The current paper thus aimed to investigate the value of two in vitro approaches to predict in vivo bioavailability of micronutrients in general and magnesium in particular, and to compare the in vivo bioavailability of two magnesium supplements with opposing in vitro results by measuring serum magnesium levels and urinary magnesium excretion’.

Point 6: The main conclusions are not supported by the data. The authors state that ‘Our study provides a valid methodology to predict in vivo outcomes and effectiveness by specific and simple in vitro approaches.’ It is completely unclear on which basis the authors draw this conclusion. Their in vitro studies showed >90% availability, whereas their in vivo studies showed no significant increase in serum Mg levels using mixed models (repeated ANOVA) analysis and only a minor increase using AUC. In my view, this shows that their in vitro tests poorly predicts in vivo outcomes.

Thank you for this critical comment. We realize that our description of the results might have been confusing, as we did not describe the significant differences of serum magnesium levels compared to pre-supplement levels (time point 0) for each condition separately, and only depicted significant differences between the different conditions. However, in Phase B of the in vivo testing, we found significant interaction effects for serum magnesium profile, both when comparing 1 tablet of Supplement A vs 1 tablet of Supplement O and 2 tablets of Supplement A vs 1 tablet of Supplement O. When performing pairwise comparisons, serum magnesium levels following ingestion of either 1 or 2 tablets of Supplement A do show statistically significant increased serum magnesium levels compared to pre-supplement levels. Since we used the highly conservative Bonferroni significance level of P<0.005, we are confident that we showed convincing and real differences in serum Mg levels between both supplements. We now also added this to the results section:

Page 12-13: “Figure 6A represents the absorption profile of 1 tablet of Supplement A vs 1 tablet of Supplement O. A significant interaction effect was present (p = 0.013) and significant different serum magnesium values between both conditions were found at 30, 90, 150, 180 minutes and 4 hours following supplement ingestion (p < 0.005, Bonferroni correction for pairwise comparison of 10 time points). Furthermore, significant higher serum magnesium levels compared to pre-supplement levels (time point 0) were found following 1 tablet of supplement A at 120, 150, 180 minutes and 4 hours following ingestion, but not at any time point following 1 tablet of Supplement O (p < 0.005, significant differences vs time point 0 are not depicted in Figure 6A). Figure 6B represents the absorption profile of 2 tablets of Supplement A vs 1 tablet of Supplement O, for which the interaction effect was also significant (p = 0.032). Significant different serum magnesium values between both conditions were found at 120, 150, 180 minutes and 4 and 6 hours following supplement ingestion (p < 0.005, Bonferroni correction for pairwise comparison of 10 time points). In accordance, significant higher serum magnesium levels compared to pre-supplement levels (time point 0) were found following 2 tablet of Supplement A at 120, 150, 180 minutes and 4 and 6 hours following ingestion (p < 0.005, significant differences vs time point 0 are not depicted in Figure 6B). No significant interaction effect was present between the absorption profile of 1 and 2 tablets of Supplement A (p = 0.179).”

We thus do believe that our conclusion is based on our statistically significant results, as the in vivo trial demonstrated that the supplement with the highest bioavailability predicted from in vitro tests, showed a significant increase in serum magnesium levels following ingestion (either 1 or 2 tablets), while the supplement with the worst predicted bioavailability does not show increased levels of serum magnesium following supplement ingestion. Thus, serum magnesium levels were significantly higher following ingestion of either 1 or 2 tablets of Supplement A compared to Supplement O, as depicted on Figure 6A & B.

Point 7: The discussion of the manuscript should be more extended. What do the authors propose? Which supplement should be used? Why? It is difficult to understand why a mix of organic/inorganic Mg would have a better bioavailability than organic mg salts.Thanks for these suggestions. However, the aim of this paper was to demonstrate that there are large differences in bioavailability of different magnesium formulations, and that in vitro bioavailability testing can predict in vivo outcomes, rather than stating which supplement should be used. Our results suggest that combined organic and inorganic magnesium is better towards bioavailability compared to inorganic magnesium, and that the solubility of a magnesium supplement is of greater relevance for in vivo bioavailability than the loading (as described in the discussion). However, we did not compare the bioavailability of a tablet containing both organic and inorganic magnesium to an organic magnesium formulation. This should be further elaborated in future research.

Following paragraph was therefore added to the discussion (line 468-476):

Based on the in vitro tests, some supplements containing organic magnesium formulations (such as Supplement B, F, H) show similar bioavailability and dissolution results compared to Supplement A, which contains both organic and inorganic magnesium. It should be noted that Supplements B, F and H contain lower magnesium content and that similar percentages of bioaccessible or bioavailable Mg as reported in Figures 1 and 2 does thus not necessarily correspond to similar bioaccessible or bioavailable amounts of magnesium, respectively. Furthermore, in the current study, no organic magnesium supplement was tested in vivo. It thus remains to be established whether in vivo bioavailability of organic magnesium formulations differs from supplements containing both organic and inorganic magnesium.

 Minor points

 Point 8: The figure quality is rather poor. Better use professional software instead of excel for figure formatting.

 Thanks for this suggestion. The figures are added to this submission in tiff format with a higher resolution, which should match the standards set by the journal.

 Point 9: Please provide quantity units (+ dimension) in all axis labels e.g. Mg concentration (mmol/L)

This is adjusted in all figures.

Point 10: Harmonize fond size, axis labels etc. in figures (remove titles above the figure)

This is adjusted and titles of figures are removed.

 Reviewer 2 Report

Interesting paper on Mg bioavailability. Data on urinary Mg excretion should be given – also in the methods section. Fig 3 can be omitted, same data as Tab. 2.

Author Response

Response to Reviewer 2 Comments

We would like to thank the reviewer for his helpful and valuable suggestions and comments, which we feel have positively contributed to the revised version of the manuscript.

Interesting paper on Mg bioavailability.

Data on urinary Mg excretion should be given – also in the methods section.

We added the description of urine collection in the method section and shortly described the results on this parameter. We chose to not give figures on urinary data as no differences in urinary magnesium excretion were present following the intake of different magnesium supplements.

Methods:

‘Serum tubes were left at room temperature for 30 minutes to obtain complete coagulation, followed by 5 minutes of centrifugation at 16000 g. Serum was then frozen at -20°C until subsequent analysis. Urine was collected in a container during the first 6 hours following supplement intake, and in the subsequent 18 hours in another container. Urine aliquots were collected from each container and frozen at -20°C until subsequent analysis. Immediately after the blood sample at 180 minutes, subjects received a standardized lunch (white sandwich with cheese, tomato, cucumber, carrots, salad and mayonnaise). During the test days, subjects were allowed to drink magnesium-free water ad libitum.’

Results:

Phase A:

‘No difference was found in urinary magnesium excretion in the first 6 hours following supplement ingestion (placebo: 0.23 mmol/h; 2 tablets of Supplement A: 0.22 mmol/h), neither in the subsequent 18 hours (placebo: 0.17 mmol/h; 2 tablets of Supplement A: 0.20 mmol/h).’

 Phase B:

No difference was found in urinary magnesium excretion in the first 6 hours following supplement ingestion (1 tablet of Supplement A: 0.17 mmol/h; 2 tablets of Supplement A: 0.16 mmol/h; 1 tablet of Supplement O: 0.17 mmol/h), neither in the subsequent 18 hours (1 tablet of Supplement A: 0.18 mmol/h; 2 tablets of Supplement A: 0.19 mmol/h; 1 tablet of Supplement O: 0.18 mmol/h).

 Fig 3 can be omitted, same data as Tab. 2.

This paper describes two different in vitro approaches to compare commercial magnesium products: the SHIME model and the dissolution tests. The in vitro SHIME experiments and in vitro dissolution tests showed a very wide variation in dissolution and absorption of the 15 tested magnesium products. Interestingly, poor bioaccessibility and bioavailability in the SHIME model clearly translates in poor dissolution, as indicated by the strong correlation between the results of both approaches which is depicted in Figure 3. Table 2 however only gives results on in vitro dissolution rates (time to release 80% of Mg content and % release of Mg after 120 min). The Mg released in pH 6.8 (last column of Table 2) was correlated with the amount of Mg bioavailable/bioaccessible (Figure 1 & 2) from the in vitro SHIME experiments and this gives extra information on the relation between the two in vitro approaches. We therefore believe that Figure 3 has an added value and should not be omitted. 

Reviewer 3 Report

I like to thank the authors for sharing their work.
It is important to be able to understand the uptake kinetics of magnesium based supplementation. Still, I have some suggestions to improve the content of the manuscript.INTRODUCTION

Use the correct name for RDA: recommended daily allowance and not Daily Recommended Allowance such as in line 43. Add after this the abbreviation (RDA) and make sure you write this correct throughout the whole manuscript. And consider referring to the actual recommendation as: Adequate Intake (AI), as average estimated requirements cannot be derived for adults.

I think it is slightly more complex then stating that 50% of the population has a lower intake than the AI. This is regionally bound and results are certainly dependent of the assessment method and the way nutrient values were calculated. I suggest to expand on this section (line 45-46) and make sure that the reader appreciates that this may differ per country (not much issues with magnesium in Germany and the Netherlands for example, but huge problems in the UK) and also discuss that within countries this may differ between populations (old vs young, sedentary vs athletes). I suggest to have a look at work published by Mensink in BJN in 2012 doi:10.1017/S000711451200565X and work from Wardenaar with athletes published in Nutrients in 2017 doi:10.3390/nu9020142.

The section containing lines 51-58 provides the opportunity to include the correct definitions for AI and their limitations (adding to my earlier comment).

Line 59-60. You can add a reference backing up what the actual popularity of magnesium is. There are a lot of studies reporting self-reported dietary supplement use.

What do you define as popular (line 60)? Wardenaar et al. (2016) doi:10.3390/sports4020033 reported a 7% self-reporting in a representative sample of the Dutch population. In Dutch athletes reporting of Magnesium was higher ~25% (Wardenaar et al. 2017, https://doi.org/10.1123/ijsnem.2016-0157). 

MATERIALS AND METHODS

Regarding the in vitro testing how did you control for the body magnesium status of your participants? Do you had any rationale for the needed number of participants in this study to obtain enough power to backup your conclusions?

Please explain why you choose to go with supplement A in the  methods.

RESULTS
Figure 1 and 2. I suggest to add dotted vertical lines to the figure separating the supplements for high, average or low absorption.
(maybe something for discussion, for practical application: dosage is very different, do you suggest that people keep this in mind when supplementing for example a very low dosed but highly bio-available supplement and vise versa?) 

Is there any chance to combine the results as described for bot experiments as what can be seen as the most optimal formulations based on the two different conditions without getting into discussion?

Please reflect on the provided dosage of Magnesium with the supplement (392 mg/day) while the Tolerable Upper Intake Levels (ULs) for supplemental magnesium was set on 350 mg/day in the US and even a lower dose of 250 mg/day from supplements was used in the Netherlands. This relates to both experiment A and B. As it difficult to change this now, this may be a discussion point. It is good to see that you also use only one tablet of product A, but the dosage of some of the supplements in general is exceeding the UL which may be a concern on the long term. And as readers may not be able to draw the correct conclusions, in relation with the actual health and safety recommendations I stress that you need to include remarks in both your conclusion of the abstract, the conclusion and the discussion about these results, because otherwise the results will be misinterpreted (as in much is better).

In-vitro testing

Line 256-258: What can be seen as slow release (add duration >120 min), and there are some supplements with a specific shorter duration then slow. 

Table 2. How relevant is the result of mg Mg released after 120 min? I understand that this is part of the model, but do you have for example also access to analysis after 30, 60 and 90 minutes? If so I suggest to add these results as depending on the feeding status of the participant this may be of relevance in relation to practical application. In relation to the discussion, you may discuss how fast the magnesium may travel through the stomach and the intestine. This also relates to your Figure 3. I assume that the incubation time between fed vs unfed conditions was the same in your model, as passage time in real time may be affected by stomach content. Do you think that this may have affected your conclusions? So why are you only showing in figure 3 the fed conditions and not the unfed conditions? It would be an easy addition to the figure (from 2 to 4 small plots).

 DISCUSSION

The concerns brought up earlier throughout my feedback should be addressed to the discussion were appropriate (if not added to the methods/results). No further feedback on the current content of the discussion which is well written, but some discussion about the applications of the results in practice is missing. Although the study is a proof of concept, people will read between the lines and may misinterpret the results, therefore discussion about the previous points is needed.

Author Response

Response to Reviewer 3 Comments

We would like to thank the reviewer for his helpful and valuable suggestions and comments, which we feel have positively contributed to the revised version of the manuscript.

I like to thank the authors for sharing their work.

It is important to be able to understand the uptake kinetics of magnesium based supplementation. Still, I have some suggestions to improve the content of the manuscript.

INTRODUCTION

Use the correct name for RDA: recommended daily allowance and not Daily Recommended Allowance such as in line 43. Add after this the abbreviation (RDA) and make sure you write this correct throughout the whole manuscript. And consider referring to the actual recommendation as: Adequate Intake (AI), as average estimated requirements cannot be derived for adults.

Thank you for this suggestion. We prefer to keep the RDA term to be consistent with what is most described in literature. We corrected this throughout the manuscript.

I think it is slightly more complex then stating that 50% of the population has a lower intake than the AI. This is regionally bound and results are certainly dependent of the assessment method and the way nutrient values were calculated. I suggest to expand on this section (line 45-46) and make sure that the reader appreciates that this may differ per country (not much issues with magnesium in Germany and the Netherlands for example, but huge problems in the UK) and also discuss that within countries this may differ between populations (old vs young, sedentary vs athletes). I suggest to have a look at work published by Mensink in BJN in 2012 doi:10.1017/S000711451200565X and work from Wardenaar with athletes published in Nutrients in 2017 doi:10.3390/nu9020142.

We thank the reviewer for this helpful suggestion. It is indeed true that this fact was quite simply described in our introduction, and that some nuance should be added to this section.

This section of the introduction in now expanded and references are added:

Line 44-48: “This is most likely a result of increased consumption of processed foods. Based on these findings, it is often suggested that over 50% of the normal population may have marginal magnesium deficiencies [7,8], although it should be noted that intake of dietary magnesium greatly differs between different countries and between different subpopulations (young vs old, sedentary vs athlete) [9,10]

The section containing lines 51-58 provides the opportunity to include the correct definitions for AI and their limitations (adding to my earlier comment).

In our opinion, we already addressed the main limitation of this aspect, namely that a lower intake than the RDA/AI does not necessarily lead to an impaired magnesium status, and that the latter should be measured to be sure whether supplementation is necessary/useful. We would prefer to not further elaborate on this in the introduction since it is not the main factor that is addressed in this manuscript and it would break the flow of the introduction. 

Line 59-60. You can add a reference backing up what the actual popularity of magnesium is. There are a lot of studies reporting self-reported dietary supplement use.

What do you define as popular (line 60)? Wardenaar et al. (2016) doi:10.3390/sports4020033 reported a 7% self-reporting in a representative sample of the Dutch population. In Dutch athletes reporting of Magnesium was higher ~25% (Wardenaar et al. 2017, https://doi.org/10.1123/ijsnem.2016-0157). 

Thanks for this suggestion. The following sentences and references were added to substantiate the popularity of magnesium supplements:

Line 61-63: “As magnesium plays an important role in physiological processes relevant to both health and sports performance, it is nowadays a popular nutritional supplement, with self-reported dietary magnesium use ranging around 7% in general population [15], and up to 25% in athletic populations [16].”

MATERIALS AND METHODS

Regarding the in vitro testing how did you control for the body magnesium status of your participants? Do you had any rationale for the needed number of participants in this study to obtain enough power to backup your conclusions?

We suppose you refer to the in vivo testing instead of in vitro testing in this comment. It is indeed true that we did not control for magnesium status during the in vivo experiments as this can not be easily determined. One possible alternative was the use of a ‘saturation phase’ during which magnesium supplements are taken a few days before the actual bioavailability testing takes place. However, as we use a cross-over design, subjects are used as their own control, and therefore the potential confounding effect of magnesium status is partly excluded.

Concerning the number of participants, no power calculation was done for in vivo phase A, as the bioavailability of Supplement A was not yet investigated and we could thus not estimate the effects on serum magnesium levels. However, in phase A, the main aim was to test our biological sampling strategy, rather than finding differences or draw conclusions from these data. Based on the results of Phase A, a power calculation was done to have an idea on the number of subjects needed in Phase B and 30 subjects were therefore chosen. Based on the results and the clear statistically significant differences (even with applying the Bonferroni correction for multiple comparisons), we are convinced that we have obtained enough power to back-up the conclusions.

Please explain why you choose to go with supplement A in the  methods.

Supplement A was the supplement with best ‘overall’ in vitro results. Both for the bioaccessible and bioavailbale (SHIME) measures, as well as for the dissolution results, supplement A belong to the three best performing supplements (highest relative absorption/release rates compared to the other supplements).

Following sentence was added to the method section to justify this choice (line 145-146): “Supplement A was chosen because it showed the best overall results in the in vitro SHIME and dissolution testing.”

RESULTS
Figure 1 and 2. I suggest to add dotted vertical lines to the figure separating the supplements for high, average or low absorption.

(maybe something for discussion, for practical application: dosage is very different, do you suggest that people keep this in mind when supplementing for example a very low dosed but highly bio-available supplement and vise versa?) 

We believe that, adding dotted vertical lines to figures 1 and 2 would add a subjective representation of high, average or low absorption, which we do not want to induce in the results section as it should only give objective results. Since it is very difficult to ‘draw the line’ between high, average or low absorption (when is it high, when it is low…?) we thus rather prefer to not add this. The aim of these figures are rather to stress out that there is a large difference in in vitro bioavailability test results of different commercially available magnesium supplements, which could have effects on in vivo bioavailability (as tested in the in vivo phase A and B).

We agree on the importance of dosing. We already elaborated on this point in the discussion, stating that solubility of a magnesium supplement seems to be of greater relevance compared to loading:

Line 456-457: “These findings suggest that the solubility of a magnesium supplement is of greater relevance for in vivo bioavailability than the loading (amount elemental magnesium).”

We also further elaborated on this by adding the following section to the discussion (line 468-476):

Based on the in vitro tests, some supplements containing organic magnesium formulations (such as Supplement B, F, H) show similar bioavailability and dissolution results compared to Supplement A, which contains both organic and inorganic magnesium. It should be noted that Supplements B, F and H contain lower magnesium content and that similar percentages of bioaccessible or bioavailable Mg as reported in Figures 1 and 2 does thus not necessarily correspond to similar bioaccessible or bioavailable amounts of magnesium, respectively. Furthermore, in the current study, no organic magnesium supplement was tested in vivo. It thus remains to be established whether in vivo bioavailability of organic magnesium formulations differs from supplements containing both organic and inorganic magnesium.

Is there any chance to combine the results as described for both experiments as what can be seen as the most optimal formulations based on the two different conditions without getting into discussion?

It is not very clear to us whether you mean the results of both SHIME experiments (Figure 1 and 2) or the results of the SHIME experiment and the dissolution tests, but we suppose you are referring to the second option. However, the aim of this paper was to demonstrate that there are large differences in bioavailability of different magnesium formulations, and that in vitro bioavailability testing can predict in vivo outcomes, rather than stating which supplement should be used.

Following sentence was added to the results to shortly elaborate on this point (line 285-289):

Taken together the results on the in vitro experiments, magnesium supplements partially or completely consisting of organic magnesium formulation such as Supplement A, B, F, H and J show the highest rate of both bioaccessibility, bioavailability and dissolution, although this was not found for the organic magnesium Supplements E and M, indicating that the magnesium formulation is not the only determining factor.

Please reflect on the provided dosage of Magnesium with the supplement (392 mg/day) while the Tolerable Upper Intake Levels (ULs) for supplemental magnesium was set on 350 mg/day in the US and even a lower dose of 250 mg/day from supplements was used in the Netherlands. This relates to both experiment A and B. As it difficult to change this now, this may be a discussion point. It is good to see that you also use only one tablet of product A, but the dosage of some of the supplements in general is exceeding the UL which may be a concern on the long term. And as readers may not be able to draw the correct conclusions, in relation with the actual health and safety recommendations I stress that you need to include remarks in both your conclusion of the abstract, the conclusion and the discussion about these results, because otherwise the results will be misinterpreted (as in much is better).

We agree that this is an important point that should be addressed in the paper. As we do not describe any dosages in the abstract, we do not think it should be mentioned at this point. Furthermore, we already elaborated on the point that solubility of a magnesium formulations is of greater importance than the loading, indicating that more is not better. In addition, following section was added to the manuscript to further elaborate on this (line 457-167):

One important notice is that the Tolerable Upper Intake Levels (UL) for supplemental magnesium are set at 350mg per day in the US and 250mg per day in Europe. It should be noted, however, that some commercially available magnesium formulations tested in the current study contain higher amounts of elemental magnesium in one tablet (see Table 1). Thus, they theoretically exceed the UL –if absorption efficiency would be 100%- and do not comply with safety and health recommendations. As in vivo bioavailability of 1 tablet of Supplement A (196mg magnesium) was not higher compared to ingestion of two tablets (392mg magnesium), and the ingestion of 2 tablets exceeds the UL, ingestion of only one tablet should be recommended. For Supplement O, containing 450mg magnesium per tablet, the UL is thus exceeded, but it is highly unlikely that this effectively led to an overdose, as no bioavailable magnesium could be measured in serum following one acute ingestion.

In-vitro testing

Line 256-258: What can be seen as slow release (add duration >120 min), and there are some supplements with a specific shorter duration then slow. 

It is indeed true that there is a range in the release rate between<10 min="" or="">120 min to release 80% of the magnesium content. This is described in the following paragraph of the result section (line 266-268):

“The release rate of magnesium from the dosage forms tested in this study ranged from immediate release (with >80% of the magnesium content released within 10 min) to slow release  (>120 min to release 80% of the magnesium content) (Table 2).”

Table 2. How relevant is the result of mg Mg released after 120 min? I understand that this is part of the model, but do you have for example also access to analysis after 30, 60 and 90 minutes? If so I suggest to add these results as depending on the feeding status of the participant this may be of relevance in relation to practical application. In relation to the discussion, you may discuss how fast the magnesium may travel through the stomach and the intestine. This also relates to your Figure 3. I assume that the incubation time between fed vs unfed conditions was the same in your model, as passage time in real time may be affected by stomach content. Do you think that this may have affected your conclusions? So why are you only showing in figure 3 the fed conditions and not the unfed conditions? It would be an easy addition to the figure (from 2 to 4 small plots).

We agree that data on other time points than 120min may be of relevance, but we do not have access to analysis done after 30, 60 and 90 minutes for the dissolution tests. It is indeed true that passage time in real time may be affected by stomach content. It is generally known that absorption starts around 1 hour following magnesium intake, tends to flatten after 2 hours and decreases from 4h after magnesium intake (Hardwick et al., 1990).

Following paragraph was added to the discussion concerning this time course in relation to the duration of blood collection in the in vivo Phase A and B of the current study (line 477-481):

“One limit of the current study is that serum magnesium levels were collected up to 6h following supplement ingestion. Magnesium is primarily absorbed in the distal parts of the small intestine and in the colon. Although magnesium absorption is known to decrease from 4-5h post ingestion, it can not fully be excluded that a minor part of the ingested magnesium might not yet be absorbed within the 6h timeframe during which serum was collected.”

We only show two correlations figures to keep some simplicity and to avoid duplicate figures that do not add extra information for the reader. Correlations with unfed conditions were also performed and statistically significant, as described in the results section (line 283-284).

DISCUSSION

The concerns brought up earlier throughout my feedback should be addressed to the discussion were appropriate (if not added to the methods/results). No further feedback on the current content of the discussion which is well written, but some discussion about the applications of the results in practice is missing. Although the study is a proof of concept, people will read between the lines and may misinterpret the results, therefore discussion about the previous points is needed.

 The discussion was extended were appropriate as already described above. Furthermore, also following section was added tp the discussion:

“Based on the in vitro tests, some supplements containing organic magnesium formulations (such as Supplement B, F, H) show similar bioavailability and dissolution results compared to Supplement A, which contains both organic and inorganic magnesium. It should be noted that Supplements B, F and H contain lower magnesium content and that similar percentages of bioaccessible or bioavailable Mg as reported in Figures 1 and 2 does thus not necessarily correspond to similar bioaccessible or bioavailable amounts of magnesium, respectively. Furthermore, in the current study, no organic magnesium supplement was tested in vivo. It thus remains to be established whether in vivo bioavailability of organic magnesium formulations differs from supplements containing both organic and inorganic magnesium.”

Round  2

Reviewer 1 Report

Although the authors provide additional explanation of their study design and interpretation of the study, the main issues remain unsolved. 

In short:

- the design is still not correct to validate the in vitro tests, thus the aim (although now changed) can still not be met by the current set of experiments

- The AUC is the only significant result, all others are not. This is a very poor result given that the best (A) vs. the worst (O) supplements from the in vitro assays were tested in vivo. Therefore the strong conclusion that in vitro approaches can predict in vivo outcomes is an over-interpretation of the results and not supported by the data. Moreover, the limited number of supplements that were tested in vivo (still, why not H?) will never be sufficient to make such a general conclusion.

- The short duration 6h and meal will interfere with the results, urinary excretion is not measured, which remain design issues.

- Despite my comments, the figures are still made with excel and of poor quality. The designs are not harmonized (presence of ticks, font size, thickness of lines etc)